# pyPAGE: A framework for Addressing biases in gene-set enrichment analysis—A case study on Alzheimer's disease

**Artemy Bakulin**[1], **Noam B. Teyssier**[2], **Martin Kampmann**[2,3], **Matvei Khoroshkin**[3,4,5,6]*, **Hani Goodarzi** [3,4,5,6,7]*

**1** Faculty of Bioengineering and Bioinformatics, Lomonosov Moscow State University, Moscow, Russia, **2** Institute for Neurodegenerative Diseases, University of California San Francisco, California, United States of America, **3** Department of Biochemistry and Biophysics, University of California San Francisco, San Francisco, California, United States of America, **4** Department of Urology, University of California San Francisco, San Francisco, California, United States of America, **5** Helen Diller Family Comprehensive Cancer Center, University of California San Francisco, San Francisco, California, United States of America, **6** Bakar Computational Health Sciences Institute, University of California San Francisco, San Francisco, California, United States of America, **7** Arc Institute, Palo Alto, California, United States of America

* matvei.khoroshkin@ucsf.edu; hani.goodarzi@ucsf.edu

**Data Availability Statement:** MSBB and ROSMAP data can be found in Synapse database and will not be shared with this paper. This data is individual level and therefore in compliance with Sage

## Abstract

Inferring the driving regulatory programs from comparative analysis of gene expression data is a cornerstone of systems biology. Many computational frameworks were developed to address this problem, including our iPAGE (**i**nformation-theoretic **P**athway **A**nalysis of **G**ene **E**xpression) toolset that uses information theory to detect non-random patterns of expression associated with given pathways or regulons. Our recent observations, however, indicate that existing approaches are susceptible to the technical biases that are inherent to most real world annotations. To address this, we have extended our information-theoretic framework to account for specific biases and artifacts in biological networks using the concept of conditional information. To showcase pyPAGE, we performed a comprehensive analysis of regulatory perturbations that underlie the molecular etiology of Alzheimer's disease (AD). pyPAGE successfully recapitulated several known AD-associated gene expression programs. We also discovered several additional regulons whose differential activity is significantly associated with AD. We further explored how these regulators relate to pathological processes in AD through cell-type specific analysis of single cell and spatial gene expression datasets. Our findings showcase the utility of pyPAGE as a precise and reliable biomarker discovery in complex diseases such as Alzheimer's disease.

## Author summary

Biological regulation is governed by a complex network of interactions involving transcription factors, RNA-binding proteins, and microRNAs. To reveal the regulatory programs underlying gene expression modulations, researchers often take advantage of gene-set enrichment analysis, an approach that studies concerted changes in a group of genes

Bionetworks policy is under controlled access. To get an access to it a Data Use Certificate should be submitted to the AD knowledge portal with the description of a study plan. Any questions regarding data access can be addressed to the help desk act@synapse.org. All results of our analysis including processed eCLIP data are available in a supplement. The program code is deposited at https://github.com/goodarzilab/pypage.

**Funding:** The work was supported by a number of grants for neurodegenerative research received by MK. These include NIH grants R01 AG062359, U54 NS123746, U01 AG072464 and a Chan Zuckerberg Initiative Ben Barres Early Career Acceleration Award. The funders had no role in study design, data collection and analysis, decision to publish, or preparation of the manuscript.

**Competing interests:** The authors have declared that no competing interests exist.

rather than observing genes in isolation. Previously, we developed a tool called iPAGE to facilitate this analysis. However, both iPAGE and other similar tools implicitly assume that different genes have uniform gene-set membership. Our recent observations challenge this assumption, revealing that some genes form regulatory networks far more frequently than others. These technical biases and redundancies in gene-set annotations complicate the accurate inference of true regulatory relationships in specific contexts. To overcome these limitations, we introduced pyPAGE, an enhanced method that extends our information-theoretic framework by incorporating conditional mutual information to account for specific biases and artifacts.

We applied pyPAGE to Alzheimer's disease, a neurodegenerative disorder marked by its complexity and characterized by progressive cognitive decline, memory loss, and behavioral changes. Pathological processes of Alzheimer's disease involve various cell types and diverse molecular mechanisms, which pose significant challenges for study. In our study we took advantage of our newly developed framework pyPAGE, performing analysis of gene expression changes at both tissue and cell type levels. This way we were able to describe regulation patterns of known regulators of Alzheimer's as well as several new ones. Additionally, we performed a cohort study of the association between the enrichment of the identified regulons and survival prognosis for patients, showing that increased activity of a subset of RBPs is positively associated with a lifespan. In total the results highlight the utility of pyPAGE and provide a valuable set of biomarkers for Alzheimer's disease.

## Introduction

Over the past two decades, gene expression analysis has emerged as an effective approach for defining cell states at a molecular level. For example, disease signature genes, defined as genes that are up or down-regulated in diseased cells, have long been used as a proxy for pathological cell identity [1,2]. However, even in the early days, it was clear that for complex diseases like cancer, signature genes fail to reproduce in independent studies [3]. This was not surprising, as it is the collective disruptions in cellular pathways and processes, rather than dysregulated expression of individual genes themselves that underlie cellular pathologies. This understanding gave rise to a suite of computational analyses that perform pathway-level or gene-set-level comparisons between samples. Broadly defined, gene-sets include (i) pathways or complexes, sets of functionally related genes that act together or sequentially, and (ii) regulons, genes that are downstream of a given regulatory factor. The latter provides an opportunity to assess the differential activity of master regulators across conditions [4]. Regulons are typically annotated using experimental data (such as ChIP-seq [5] and CLIP-seq [6]) or inferred using sequence binding preferences of each regulator (AlignACE [7,8], FIRE [9], DeepBind [10], DeepSEA [11], DanQ [12]). We have previously shown that application of this methodology to large scale gene expression datasets reveals the underlying regulatory programs that are key to the emergence or maintenance of a given cellular state [1]. However, as we will discuss here, additional challenges needed to be addressed to further improve our ability to detect perturbations in regulatory programs.

There are myriad tools implementing pathway level analysis; Garcia-Campos et al [13] categorize these tools into two main categories: over-representation analysis (ORA) methods and Functional Class Scoring (FCS) methods. ORA relies on the concept that the top-scoring genes should carry the highest weight; therefore, the ORA methods operate under the null hypothesis

that if a number of differentially expressed genes is chosen, the fraction of genes that belong to any given pathway does not exceed the fraction expected by chance. In contrast, FCS methods operate under the assumption that pathways might consist of many genes that show heterogeneous yet coordinated changes in expression. Such methods make use of all available measurements and often do not rely on arbitrary thresholding.

In these methodologies, pathway annotation databases, such as Gene Ontology [14,15] and MSigDB [3,16], can be used to represent pathways as gene-sets or binary vectors where a gene can either belong to a given pathway, or not. However, this commonly used representation of pathway membership creates an intrinsic bias for several reasons. First, genes are multifunctional and some genes are far more studied than others and therefore are simultaneously annotated as part of many pathways. Second, the more abundant a given protein, the more often it is detected in various assays (such as affinity purification); therefore, abundant proteins similarly tend to be categorized as members of many complexes and gene-sets. Such biases can have a broad impact on target selection and pathway identification in large-scale datasets. For instance, identifying a protein involved in only a few gene-sets as differentially expressed narrows the range of pathways considered relevant, compared to a similar finding about a protein associated with many gene-sets. To our knowledge, no available pathway-analytical tool explicitly takes such biases into account.

Here, we propose a pathway analytical approach that explicitly models the representation bias present in gene-set annotations. This framework is an extension of our iPAGE package developed more than a decade ago [1]. In iPAGEv1.0 we used mutual information to directly estimate the dependency between expression values and gene-set annotations. In contrast, pyPAGE relies on calculating conditional mutual information, which allows for explicit consideration of annotation bias as a covariate. Here, we provide a number of benchmarks showcasing the utility of pyPAGE as a robust pathway analytical tool that surpasses its predecessors. We demonstrate that pyPAGE can effectively reveal the regulatory mechanisms underlying biological observations, which is a crucial first step towards a molecular understanding of their origin and maintenance.

The generalizability of pyPAGE allows it to infer regulatory changes at various molecular levels using a variety of different data modalities, including differential gene expression and differential stability estimates inferred from both bulk and single cell data. We took advantage of this to carry out a comprehensive analysis of gene expression programs in Alzheimer's disease. While AD is currently characterized at molecular, histological, and symptom levels [17], the regulatory networks, both transcriptional and post-transcriptional, that accompany and likely drive cellular pathogenesis in AD are not fully explored. Leveraging pyPAGE, we have generated novel insights into the role of key master regulators, including transcription factors (TF), RNA binding proteins (RBP), and microRNAs (miRNAs), in shaping the pathological transcriptome in AD. First, we perform an analysis of bulk RNA-sequencing data from Mount Sinai Brain Bank (MSBB) [18], a consortium that holds a large collection of characterized brain data from both healthy donors and patients with pathological conditions. We focus on the analysis of samples of Brodmann Area 36, since it is one of the most vulnerable brain regions in AD [19] and we aimed at identifying a core set of deregulated regulons that are inherent to the disease. Next we provide a cell-type specific characterization of differential activity of the identified regulons using two single-cell RNA sequencing datasets of human prefrontal cortex [20,21], followed by description of their spatial regulation patterns inferred from spatial transcriptomics data of middle temporal gyrus [22]. Though we analyze different brain regions, we show an overlap between their regulatory signatures. Finally, on a different bulk RNA-sequencing dataset of dorsolateral prefrontal cortex we showcase the utility of the identified factors for a stratification of patients based on the activity of these factors inferred

using pyPAGE. Using pyPAGE in these analyses provides a significant advantage by effectively accounting for uncertainty around putative regulatory interactions and non-specific and artifactual hits, therefore improving our ability to capture real biological signals in the data.

## Results

### Gene-set annotations often form scale-free networks

In every annotation of biological regulons, gene-set membership varies significantly among all genes. Typically, this measure of gene-set membership reflects the number of functional interactions a gene engages in. Current analytical tools assume gene-set membership to follow a normal distribution (**Fig 1A**). However, in practice this assumption rarely holds true and gene-set membership instead follows the power law (**Fig 1A**). We have demonstrated this across multiple commonly used gene-set annotations: molecular pathways, transcription factor target genes, RNA-binding protein regulons and miRNAs targets (**S1A–S1D Fig**). In each of these annotations, the degree distribution of the gene-set membership could be closely approximated using the power law function ($R^2 > 0.9$), which typically peaks at 1 with a long distribution tail. This implies that like most biological networks, gene-set annotations also form scale-free networks.

Further analysis of these gene-set membership distributions revealed that in many cases the shape of the right tails significantly deviates from the power law (**Fig 1C**). Such deviations indicate a bias in the underlying annotations. We posit that this bias arises from the experimental and analytical techniques that are used to compile them. For example, we observed that the

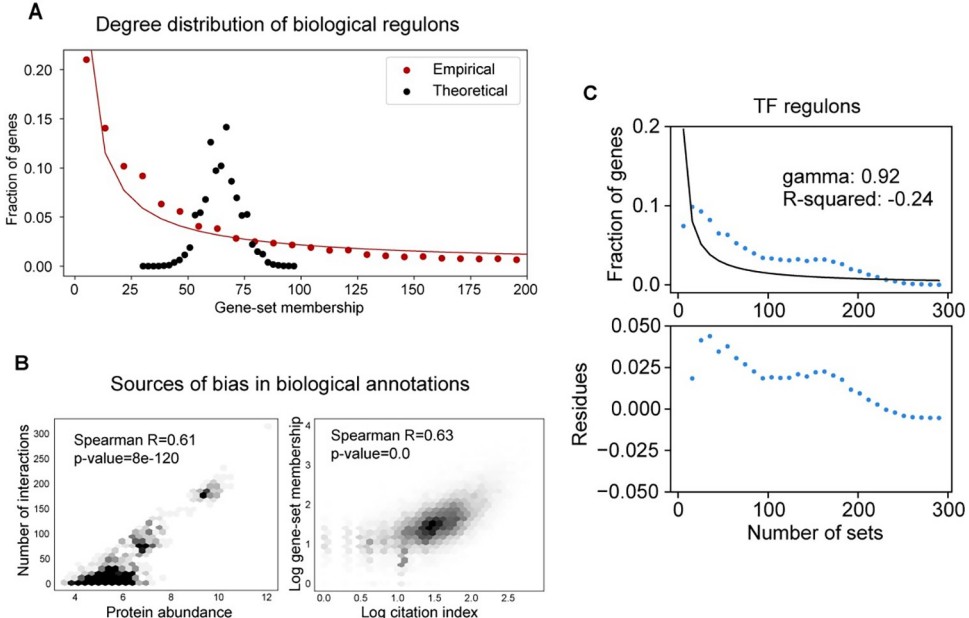

**Fig 1. Bias in gene-set annotations.** (**A**) Comparison between the theoretical and empirical degree distributions of gene-set membership in gene-set annotations. The red line represents the curve-fitting of the power law function to the observed distribution. (**B**) Scatter plots representing major sources of bias in biological annotations. The left panel represents the association between the protein abundance and the number of interactions a gene has in the STRING database. The right panel represents the association between the citation index of a particular gene and its gene-set membership in an annotation of biological pathways. For each association, we also report correlation between values. (**C**) Characteristics of gene-set membership degree distribution within TF regulon annotations are depicted. The top plot displays the observed distribution with a power law function fitted to it, including reporting the gamma parameter and the R^2 value. The bottom plot illustrates the deviation of the observed distribution from the expected power law.

more a given gene is studied the more likely it is to have multiple entries in a knowledge-based annotation (**Fig 1B, right panel**). Another source of bias may be technical limitations including low sensitivity of detecting weak functional interactions. Consistently, we have observed that, in the human interactome, the number of physical interactions for a given protein correlates with its protein abundance (**Fig 1B, left panel**). Overall, these findings indicate that there are important structural biases in the commonly used gene-set annotations. As we will demonstrate below, these inherent biases, which are overlooked in most gene-set enrichment analyses, have an impact on the insights that can be gleaned from the underlying data using current methodologies.

## pyPAGE overcomes structural annotational biases in gene-set enrichment analysis

Current gene-set enrichment analysis tools assume that every gene in a regulon is equally informative of regulon activity. However, as we demonstrated above, gene-set membership significantly varies between genes. As a consequence, differential expression of genes with specific functions provides stronger evidence for deregulation of its corresponding gene-sets than changes in expression of a more promiscuous gene. To address these biases and imbalances, we developed pyPAGE, an improved gene-set enrichment analysis method. pyPAGE is an extension of our previously published iPAGE framework [1] and can be used to analyze various gene-set annotations such as Gene Ontology, MSigDB, and regulons of TFs, RBPs and microRNAs. To correct for imbalances in gene-set membership, pyPAGE uses the concept of conditional mutual information [23] and measures the dependency between input data (e.g. differential gene expression) and a specific gene-set while taking into account the total number of gene-sets every given gene belongs to. The statistical significance of this dependency is then determined using a permutation test. Like its predecessor, pyPAGE detects any non-random patterns of enrichment or depletion of target genes and even non-monotonic complex relationships are captured (e.g. dual regulators). It also does not depend on the particular distribution of the input data and can be applied to both discrete (e.g. cluster indices) and continuous (e.g. log-fold changes) inputs.

pyPAGE allows for the comprehensive analysis of gene expression changes at multiple levels, including biological pathways as well as transcriptional and post-transcriptional regulons. To standardize the analysis of biological datasets using pyPAGE, we propose the workflow described in **Fig 2A**. We begin with initial preprocessing of RNA-seq data to obtain estimates of transcript abundance in each sample. Then, for each gene, we compute an estimate of its log fold change and a corresponding p-value using any of the available tools, such as DESeq2 [24], limma [25] or edgeR [26]. These values form the basis for an estimate of differential gene expression as demonstrated by Formula 3. Next, we apply pyPAGE to identify biological pathways and transcriptional factor regulons that are significantly associated with the differential gene expression. This approach can be broadly used for both bulk and single cell RNA-seq data, which in the latter case would allow better capture of cell-type specific patterns of regulation. Since miRNAs and RBPs impact gene expression by influencing RNA decay, differential RNA stability measurements provide a superior source of signal for detecting significant post-transcriptional regulons. To incorporate this information, we propose using our recently published tool REMBRANDTS, which infers changes in transcript stability by comparing the numbers of intronic and exonic reads [27]. These measurements can then be used to estimate the magnitude of stability changes between two groups and its significance, from which differential stability can be calculated in a similar way to differential gene expression. In addition to the comparative analysis between groups of samples, pyPAGE supports analysis of regulons in

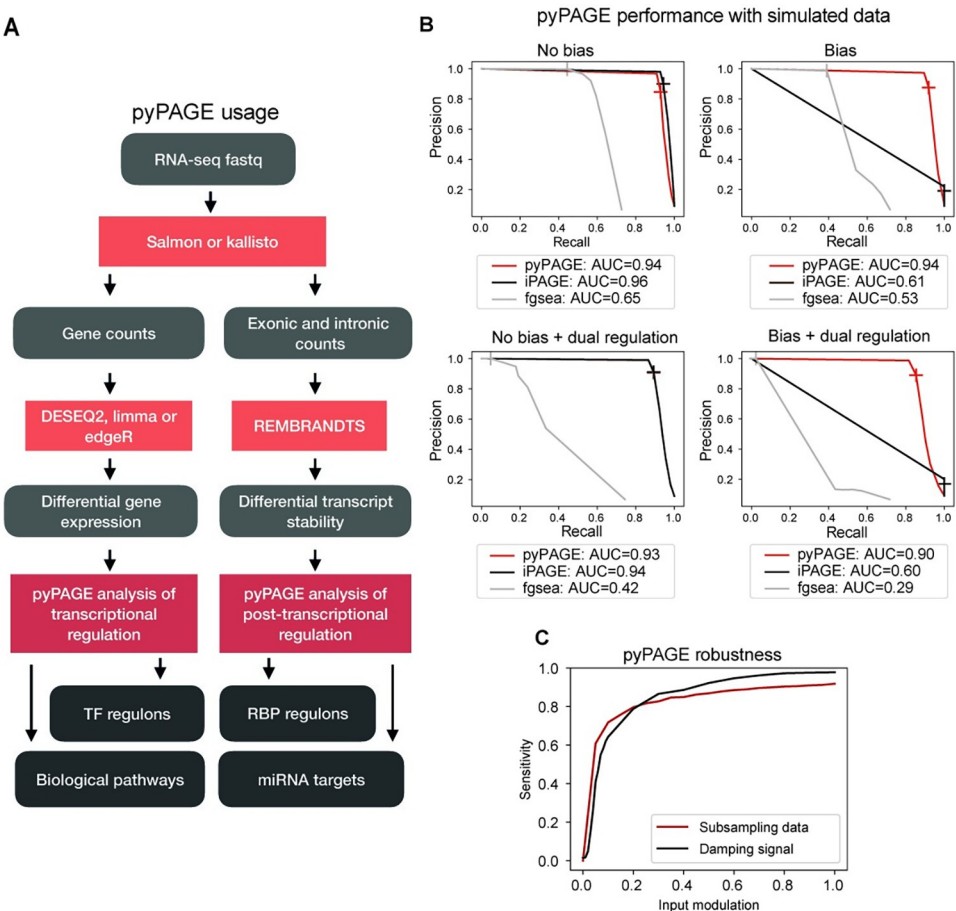

**Fig 2. pyPAGE is a novel framework for inference of differentially regulated gene-sets. (A)** Schematic of the pipeline we propose for the analysis of bulk RNA-seq data using pyPAGE. The pipeline starts with preprocessing of RNA-seq data and then diverges into two branches: one for the analysis of transcriptional regulation and the other for the analysis of post-transcriptional regulation. **(B)** Precision-recall curves demonstrating the performance of pyPAGE and benchmarking it against iPAGE and fgsea. The analysis was made in 4 simulated scenarios with and without added biases and with or without dual regulation patterns. As a general metric of performance we report PR-AUC score, also cross glyphs mark the performance at p-value threshold equal to 0.01. **(C)** Graphical representation of pyPAGE's robustness to variations in input data quality. The analysis incorporates two distinct curves illustrating the effects of: 1) subsampling the data from 5% to 100% in increments of 5%, and 2) adjusting the parameter that dictates the fraction of deregulated genes within each regulon (note that the default value for this parameter is 0.5 which explains divergence of two curves at 1.0).

each sample individually using expression or stability estimates that are normalized across samples.

## pyPAGE outperforms existing tools on simulated datasets

To assess the ability of pyPAGE to account for biases while maintaining a high baseline performance relative to its predecessors, we used several simulation-based validation strategies. By controlling the "ground truth" signal in simulated data and setting *a priori* expectations for gene-set modulations, we were able to measure the sensitivity of pyPAGE and compare its performance to other similar tools.

The main objectives of these simulations were (i) to demonstrate that implicit biases in gene-set annotations can negatively impact the quality of gene-set enrichment analysis, and

(ii) to assess whether pyPAGE effectively addresses this issue and robustly identifies deregulation of gene-sets from biased annotations. To simulate pathway annotations, we randomly selected subsets of varying sizes from a set of genes. We introduced artifacts by enhancing the likelihood that certain genes would be included across all pathway annotations. Specifically, we designated a set of 'biased genes' and added a random sample from this group to each simulated pathway, as detailed in the Methods section. Our secondary goal was to demonstrate that pyPAGE can detect any non-random patterns in data, both monotonic and non-monotonic. Though activity of most factors is associated with either upregulation or downregulation of their target regulons, there are nevertheless factors that function as dual regulators simultaneously exerting both effects. Therefore, we simulated two types of datasets with monotonic and dual regulation patterns.

We assessed iPAGE [1] and fgsea [28] as comparators to pyPAGE. The performance of each method was measured by its ability to recover the artificially perturbed gene-sets in each simulated dataset. We employed four simulation scenarios to test how well each algorithm handles bias in annotations and captures monotonic and non-monotonic regulatory patterns. In each case, we report relevant performance metrics (**S2 Fig**). However, since the number of unperturbed gene-sets in the simulation is much greater than the number of perturbed gene-sets, we specifically focus on the precision-recall curves (**Fig 2B**). In these benchmarks, pyPAGE and iPAGE performed equally well predicting deregulation of gene-sets from unbiased annotations both with monotonic and non-monotonic signals (PR-AUC>0.90). However, when using biased annotations, performance of iPAGE decreased dramatically by more than 30% as indicated by changes in PR-AUC score. Performance of pyPAGE, on the other hand, remained largely unchanged. Fgsea performed worse than pyPAGE and iPAGE using the same simulated datasets across all tasks. Firstly, it failed to accept several gene-sets as input and could not analyze their deregulation because of imbalances in their input distributions. Secondly, it largely failed to capture non-monotonic and bimodal signals, which led to a 35% decrease in PR-AUC score as compared to the case with monotonic signals. Finally, annotational biases led to reductions in its PR-AUC by 18% and 30% for monotonic and non-monotonic signals, respectively. These results demonstrate that based on the criteria formulated by our two validation objectives, pyPAGE outperforms other tools in being robust to biases in gene-set annotations and identifying complex patterns in data.

## pyPAGE is both robust and sensitive in detecting dysregulated gene-sets

To further assess the robustness and sensitivity of pyPAGE, we implemented several additional simulation strategies. Our goal was to demonstrate that the identified gene-sets remained consistent across independent runs, even as signal-to-noise was reduced. To suppress the signal in the input data, we employed two approaches. First, to measure the sensitivity of pyPAGE to the strength of the input signal, we directly controlled the magnitude of the simulated gene expression changes by tuning it as a hyperparameter. In the second approach we simulated new datasets by randomly selecting and changing a subset of genes in the target gene-set to evaluate pyPAGE's robustness to incomplete data or imperfect gene-set annotations [9]. pyPAGE proved to be highly robust to the choice of simulation parameters controlling signal expressiveness within a wide range (**Fig 2C**), which is showcased by its ability to capture more than 80% of the target gene-sets even after an 80% decline in the strength of a signal controlled by the simulation parameters. Similarly, even after sub-sampling the genes to as low as 20% of the original counts, pyPAGE was still able to recover more than 80% of the perturbed target gene-sets. These results suggest a strong potential for pyPAGE to be highly useful on single-cell RNA-seq datasets, where substantially fewer genes are captured relative to bulk data.

## Comprehensive analysis of regulatory perturbations in Alzheimer's disease

As we showed above, pyPAGE provides a sensitive and robust approach for identifying the regulatory programs that drive changes in gene expression. Extending our work to real-world data, we sought to use pyPAGE to study regulatory perturbations in Alzheimer's disease (AD). This highly complex disease involves dysregulation of many cellular pathways and processes. The recent explosion in the amount of transcriptomic data from AD patients presents an opportunity to systematically investigate the transcriptional and post-transcriptional regulatory programs mediating AD–a first step toward potential new therapeutic targets.

AD primarily causes the atrophy of the large cortex, subcortex and a number of other areas of the brain [29]. Pathologically, it is characterized by neuronal loss and destruction of synaptic connections, which are linked to the accumulation of amyloid beta and tau protein. Previous research also points to a variety of other pathological mechanisms that may be involved in AD, such as immune dyshomeostasis [30], oxidative stress [31] and metal imbalance [32]. A handful of studies have sought to identify master regulators of these changes based on the regulators' differential gene expression [33–35]. However, since the activity of proteins is often controlled at the post-translational level [36], it cannot be reliably inferred from their mRNA expression levels alone. Therefore, we sought to instead infer their differential activity from changes in the expression of their downstream target genes. In our earlier work, we demonstrated that an information-theoretic approach can be effectively employed to capture regulatory programs that underlie complex human diseases [1]. pyPAGE provides the opportunity to infer associations between AD pathology and deregulation of TFs, miRNAs, and RBPs capitalizing on both known (sourced from curated databases) and putative (derived from high-throughput assays) interaction datasets (**Table 1**).

Using publicly available RNA-seq data from the MSBB study [18], we performed a comparison of samples from the BM36 brain region between individuals with an AD diagnosis and without, while controlling for biological variables such as sex. To capture differences in transcriptional control, we used our previously curated annotation of *cis*-regulatory elements and transcription factor binding sites [37]. Similarly, post-transcriptional regulatory changes were predicted based on the combined annotation of known RBP binding sites from various sources (see Methods) and miRNA recognition elements [38,39]. To ensure that the identified regulatory programs reflect AD pathogenesis and not changes in cellular composition artificially read out in bulk data, we also analyzed their cell-type specific activation patterns using single cell measurements, as well as their clinical relevance in predicting long-term survival of AD patients.

## Transcriptional regulatory modules associated with Alzheimer's disease

In our analysis of transcriptional deregulation in AD, we observed significant disease associations for 15 transcription factors (TFs) (**Fig 3A**) most of which are regulators of neuronal functions (**Fig 3E**). We also demonstrate that changes in expression of these TF regulons occur in both female and male donors (**S3A Fig**) and in each age group (**S3B Fig**). For a number of

**Table 1. Summary of pyPAGE findings.**

| | |
|---|---|
| **Transcriptional factors** | NR2F1, ATF4, TFAP2, CDUX4, SOX10, ZNF92, NFKB1, KDM5A, JARID2, FOXP2, OTX2, CDX2, SIX2, SMAD1, TCF4 |
| **RNA binding proteins** | ILF3, LIN28A, DROSHA, METAP2, PABPC4, ZFP36, HNRNPK, HNRNPD, LARP4, SERBP1, RPS3, ALKBH5, IGF2BP1, PCBP1, EWSR1, FXR1 |
| **miRNAs** | miR-106B-5P, miR-20A-5P, miR-124-3P, miR-6825-5P, miR-506-3P, miR-8067, miR-302C-5P, miR-506 (GTGCCTT) |

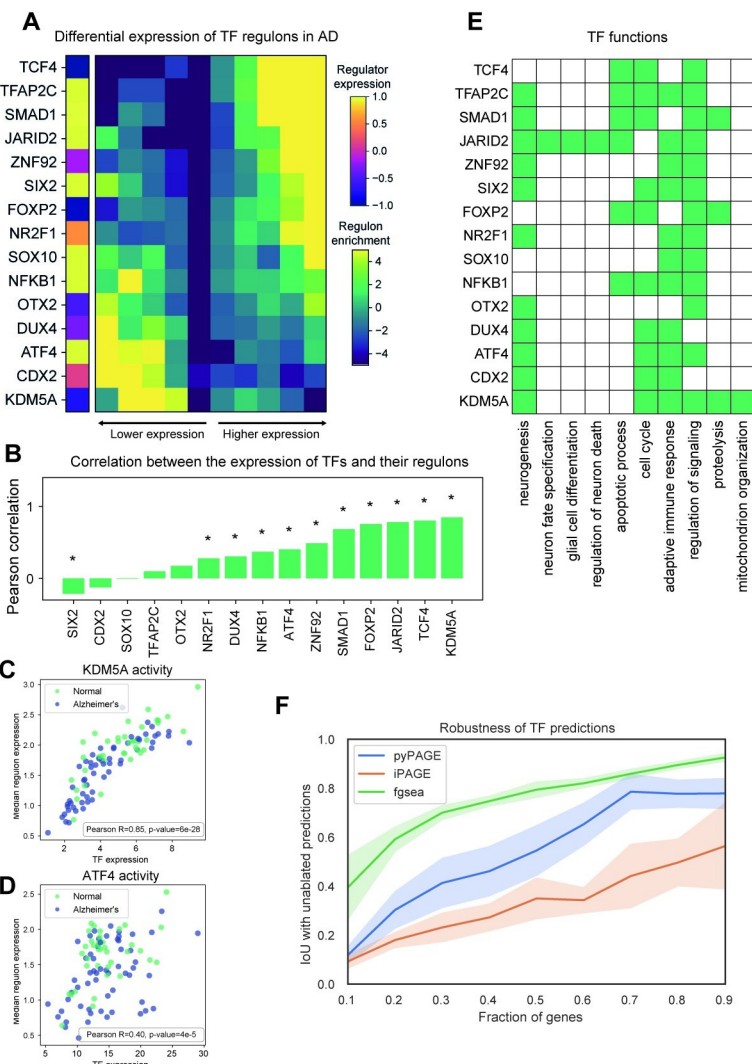

**Fig 3. Transcription factors associated with gene expression changes in Alzheimer's Disease.** **(A)** Regulons of TFs differentially expressed between AD and non-AD samples discovered by pyPAGE. In this representation the rows correspond to TFs and columns to gene bins of equal size ordered by differential expression, the cells are colored according to the enrichment of genes from regulons in a corresponding bin. The leftmost column of the heatmap depicts the differential expression of the regulator itself. **(B)** The barplot representing Pearson correlations between the expression of TFs and of their regulons, as measured by median TPM of its members. Asterix indicated significant correlation (p-value<0.05). **(C)** The scatter plot demonstrating association between the expression of the well-known AD regulator KDM5A with the expression of its regulon. **(D)** The association between the expression of another AD regulator ATF4 with the expression of its regulon. **(E)** Biological roles of the identified TFs inferred based on the functions of the genes controlled by these TFs. In these heatmap colored cells correspond to TFs whose regulons are significantly (p-value<0.05) enriched with genes from a corresponding biological pathway based on PantherDB. **(F)** Plot showcasing how robust are predictions of three different methods to subsampling of expression data. To measure consistency of the predictions we computed intersection over union (IoU) of the method's output with and without subsampling of genes.

these regulons, we also observed concomitant changes in the expression levels of their associated TFs as well (**Fig 3B**). While the correlated expression of a TF and its regulon strengthens the evidence for the observed associations, as mentioned earlier, differential expression of a TF is only one possible way that it may acquire differential activity. For example, our analysis correctly identified two well-known transcriptional regulators of AD, KDM5A [40] and ATF4

[41]–but only the mRNA expression level of KDM5A showed a high correlation with the mean expression of its regulon ($R^2 = 0.85$) (**Fig 3C**) while ATF4 did not ($R^2 = 0.40$) (**Fig 3D**). Indeed, ATF4 levels are known to be controlled at the level of translation [42].

These transcriptional signatures are robustly recovered by pyPAGE even when part of the input expression data is missing due to the subsampling of genes (**Fig 3F**). By conditioning on the regulon membership of each gene, pyPAGE avoids reporting spurious non-robust associations and outperforms iPAGE by reporting a more consistent set of genes with different fractions of the input (as indicated by higher intersection over union). Fgsea similarly shows high consistency of its predictions across runs with ablated data. However, in contrast to pyPAGE, which identifies deregulation of only 3% of all TF regulons, Fgsea detects significant associations with over half (53%) of all TFs, which is highly unlikely to be biologically meaningful. Indeed, our previous studies of the pathological hallmarks of AD, specifically tau accumulation [43] and microglia activation [40], revealed that fewer than 10 out of the top 100 genes associated with these changes were transcription factors, underscoring that AD pathology is driven by the modulation of a limited number of key factors. This suggests that many of the Fgsea hits could be false positives, in line with our observations when benchmarking against synthetic data (**Fig 2B**). Overall, the analysis highlights that pyPAGE is adept at identifying both robust and specific signals within AD datasets.

Consistent with their association with AD pathology, many of the identified TFs are key regulators of brain development and homeostasis. Systematic analysis of the functions represented among the regulons of these identified TFs (see Methods), revealed that 11 out of 15 TFs regulate molecular programs related to neurogenesis (**Figs 3E and S4 and S1 Table**). For example, ATF4 maintains excitatory properties of neurons [44], whereas the generation of oligodendrocytes [45] is controlled by SOX10. While the majority of the identified TFs function as transcriptional activators, two of them, JARID2 and KDM5A, are also involved in epigenetic reprogramming, which is a hallmark of AD pathology [46,47]. Additionally, several of the identified factors control adaptive immune response and protein metabolism, linking these regulons to other key hallmarks of AD: inflammation, amyloid accumulation and abnormal phosphorylation of tau protein.

## Cell-type specific patterns of deregulation for AD-associated TF regulons

Since the analysis of bulk data captures a mixture of cell types, it often requires major shifts in gene regulation in order to generate a discernible signal. Moreover, some of the patterns observed in bulk may reflect changes in cell-type abundance rather than bona-fide reprogramming of gene expression. To account for this possibility and improve resolution of our analysis, we applied pyPAGE to the ROSMAP single-cell RNA-seq study [20], which includes the major brain cell types in samples from both AD patients and healthy donors (**Fig 4A**). For each cell type, we estimated differential gene expression between patients and healthy controls and then applied pyPAGE to predict deregulated TF activity. Since the approach implemented by pyPAGE is independent of any particular distribution of the input data, it was seamlessly extended to the scRNA-seq data domain. The benefits of using pyPAGE over other tools developed for analysis of scRNA-seq data (e.g., Vision [48], Triangulate [49], scFAN [50]) are the same as the ones we enumerated for bulk datasets. As we showed earlier (**Fig 2C**) pyPAGE is robust to data sparsity and it accounts for biases in our annotations. As a result, we expect a reduction in the number of false positives, which makes pyPAGE especially well-suited for regulon exploration of cell-type specific gene expression patterns.

As mentioned above, the primary goal of integrating single-cell data into our AD analysis workflow was to measure contribution of the identified regulons to the gene expression of

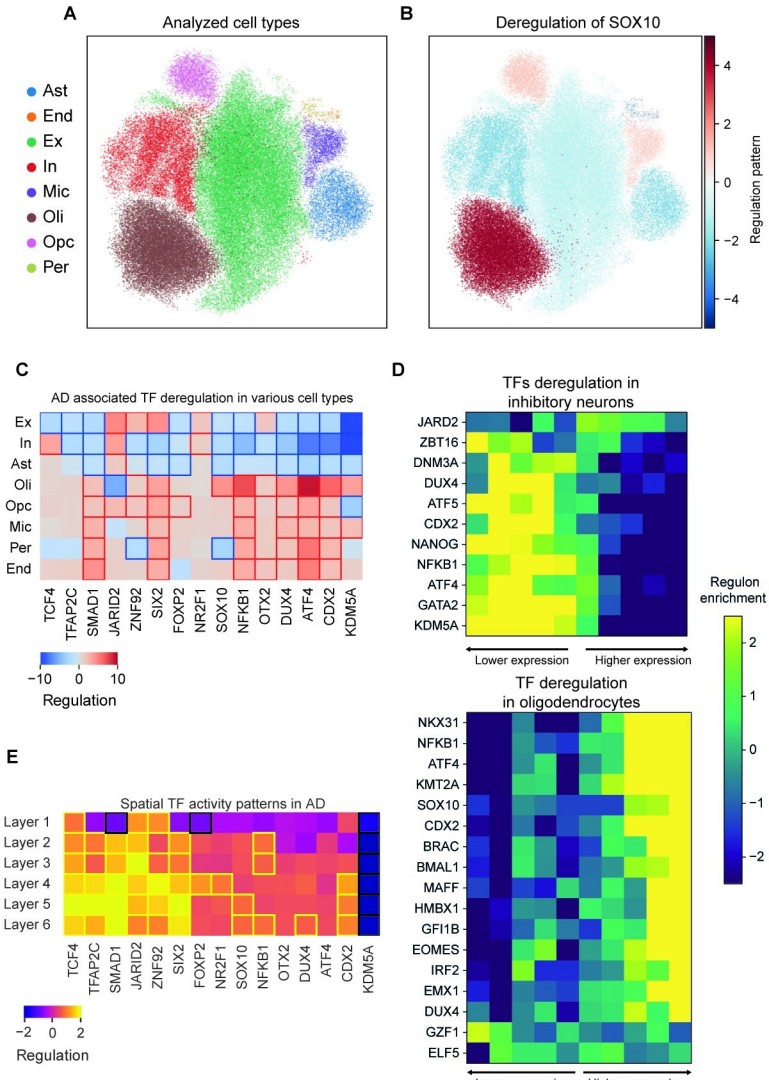

**Fig 4. Cell type and regional specific differential activity patterns of transcriptional factors in AD.** (**A**) Cells from the analyzed ROSMAP dataset represented on a force-directed graph embedding. The clusters are colored according to cell-types: excitatory neurons (Ex), inhibitory neurons (In), astrocytes (Ast), oligodendrocytes (Oli), oligodendrocyte progenitor cells (Opc), microglia (Mic), endothelial cells (End), pericytes (Per). (**B**) The same cell-type clusters colored according to differential activity of SOX10 between cells from non-AD and AD samples estimated using pyPAGE. The magnitude of the regulation pattern was calculated as scaled conditional mutual information multiplied by the factor representing the direction of deregulation. (**C**) Summary of the cell-type specific deregulation patterns of the TFs identified in the analysis of the bulk data. Heatmap cells with significant associations (p-value<0.05) are framed. The regulation is calculated as the normalized conditional mutual information of the relationship multiplied by the sign of the log fold change. (**D**) Heatmap representations of concordant expression changes in expression of TF target genes in inhibitory neurons and oligodendrocytes. Here rows correspond to TFs and columns to gene bins of equal size ordered by differential expression, the cells are colored according to the enrichment of genes from regulons in a corresponding bin. (**E**) This heatmap summarizes deregulation patterns in various cortical layers of TFs that we previously identified in the analysis of bulk data. Heatmap cells with significant associations (p-value<0.05) are framed. Regulation pattern is estimated as normalized conditional mutual information of the association multiplied by the sign of log fold change.

individual cell types. As expected, all of the 15 TF regulons we identified from bulk data showed significant deregulation in at least one cell type. Strikingly, 6 of them showed significant associations with gene expression across most of the major cell types, indicating their diverse roles in brain homeostasis. For example, ATF4 is known to act in a cell type specific

manner and depending on the cellular context, it activates both pro-survival and pro-death pathways [51]. Accordingly, in our analysis, we observed that the ATF4 regulon showed divergent patterns of activity in various cell types. SOX10 being a major regulator of oligodendrocyte survival [45] and inhibitor of astrocyte differentiation [52] showed increased activity in the former cell type and decreased activity in the latter cell type(**Fig 4B**). Or, similarly, TCF4 which is known to be a major regulator of neurogenesis [53] showed upregulation of its regulon in excitatory neurons and downregulation in inhibitory neurons. Finally, we observed strong evidence of epigenetic reprogramming, manifested either by activation or repression of chromatin modifiers KDM5A and JARID2 target regulons in all major cell types. Taken together, these findings illustrate cell type specific regulatory programs involved in AD, further highlighting the generalizability of pyPAGE to single cell transcriptomic measurements.

Next, we asked whether there were patterns in the single-cell data that had remained hidden in our analysis of bulk RNA-seq. We analyzed the cell type specific differential regulon activity of factors that were not identified in the bulk analysis (**S5A–S5E Fig** and **Table 2**). Intriguingly, inhibitory neurons showed downregulation of factors involved in neuronal differentiation (DMT3A [54], NFKB1 [55], KDM5A [56]) (**Fig 4D, top panel**) and, likewise, in excitatory neurons we observed inhibition of TF regulons that characterize mature neurons (NCOR2 [57], KDM5A [56]) (**S5B Fig**). Excitatory neurons also showed activation of regulons controlled by neuroprotective factors (SIX2 [58], NR1H3 [59], TCF7 [60]) (**S5B Fig**). Unlike inhibitory neurons, glial cells showed upregulation of developmental factors and similarly to excitatory neurons activation of cytoprotective factors, such as shown, for example, by activation of regeneration programs in oligodendrocytes (NKX3-1 [61], NFKB1 [62], EMX1 [63], IRF2 [64]) (**Fig 4D, bottom panel**). Together, these observations support the hypothesis that AD pathology disrupts gene expression programs that give rise to and maintain cellular identity in neurons [65], while simultaneously triggering cytoprotective response in glia.

## Spatial patterns of deregulation for AD-associated TF regulons

Beyond the cell type specificity of gene expression programs, the spatial patterning and physical organization of these cell types and functions within the brain can influence physiology in ways that cannot be gleaned from bulk or single cell RNA-seq data. Therefore, we extended our analysis to spatial transcriptomics data, here focusing on a Visium platform [22] dataset from the middle temporal gyrus–which is a vulnerable region in the early stage AD [66]. For

**Table 2. Summary of pyPAGE single-cell analysis of transcriptional deregulation.**

| | |
|---|---|
| **Excitatory neurons** | KDM5A, SIX2, JARID2, NR1H3, CDX2, NCOR2, NR2F1, TCF7, DUX4, NEUROD1, FOXJ2, AHR, THA11 |
| **Inhibitory neurons** | KDM5A, CDX2, ATF4, NFKB1, GATA2, JARID2, DUX4, ATF5, NANOG, DNM3A, ZBT16 |
| **Oligodendrocytes** | NKX3-1, ATF4, NFKB1, KMT2A, CDX2, SOX10, MAFF, BRAC, EOMES, GFI1B, HMBX1, EMX1, DUX4, ELF5, IRF2, GZF1, ARNTL |
| **Oligodendrocyte progenitor cells** | BRAC, FOSL2, HOXC9, ATF4, RFX1, MEF2B, DUX4, IRF2, GLI1, NR2F1, HXA13 |
| **Astrocytes** | ATF4, NANOG, GATA1, NR1H3, SOX10, CRBL2, CASZ1, HMBX1, HOXC9, SATB1, THA11 |
| **Microglia** | BATF, MAF, CDX2, BRAC, DUX4, HOXC9, OTX2, SOX10, GFI1B, MAFF, FOXA1, EOMES |
| **Pericytes** | ATF4, SIX2, NKX3-1, FOSL2, CDX2, HOXC9, NFKB1, MAFF, DUX4, NANOG, HXC6, DNM3A, GATAD2A |
| **Endotheliocytes** | ATF4, MEF2A, NKX3-1, FOXH1, ATF5, DUX4, KMT2A, HMBX1, ARID1B, DNM3A, MAFF, ZNF92, ZN335, GFI1B, FOXP2 |

each cortical layer, we computed differential gene expression between AD and control samples and further estimated TF deregulation using pyPAGE.

As expected, 13 of the TFs regulons that we previously identified as associated with AD in the bulk data showed significant deregulation in at least one cortical layer in the spatial analysis with 7 of them showing significant deregulation among all layers (**Fig 4E**). Intriguingly, many of these TFs exhibit distinct spatial patterns of activation of their corresponding regulons. For example, SOX10, DUX4, and CDX2 regulons are preferentially activated in deep cortical layers, while FOXP2 and NR2F1 regulons are active in the middle layers. While most of these associated TF regulons were activated in AD compared to control samples, the KDM5A regulon exhibits strong downregulation across all layers in AD samples. Overall, this analysis shows that transcriptional programs affected by AD have different levels of regional specificity.

## Systematic identification of RNA-binding proteins associated with pathological post-transcriptional gene regulation in AD

In the previous sections, we explored AD-associated transcriptional programs. Following, we performed a similar analysis of post-transcriptional programs controlled by RBPs and miRNAs. To do that, we computed unbiased estimates of transcript stability from MSBB RNA-seq data using our previously developed tool REMBRANDTS [27] and then assessed differential stability between AD patients and healthy donors. Noteworthy, in the analysis of post-transcriptional regulation it is critical to use estimates of RNA turnover, which is controlled by RNA-binding proteins and microRNAs, and cannot be substituted by differential gene expression, since it mainly reflects effects of transcriptional regulators. Further, we compiled and collated an annotation of post-transcriptional regulons using multiple sources: namely ENCODE [67] and POSTAR databases [68], as well as the Mireya Plass study [69]. By applying pyPAGE to this new comprehensive annotation of RBP targets we were able to disentangle the contribution of each RBP to the changes in stability and uncover novel post-transcriptional regulators associated with AD.

Utilizing the framework described above, we identified 16 candidate RBPs that are associated with differential RNA stability in AD (**Fig 5A**). Several of these proteins have been previously implicated in AD pathology. For example, in an earlier study, we had identified ZFP36 as a regulator of AD-related changes in RNA stability in an independent dataset [27]. Here, pyPAGE confirmed that ZFP36 activity is indeed perturbed in AD as evidenced by upregulation of the genes targeted by this RBP. Similarly, pyPAGE identified deregulation of HNRNPK and HNRNPD, two RBPs that are silenced in AD due to epigenetic alterations [70,71]. The HNRNPK regulon is destabilized in AD samples, indicating that this RBP acts as an enhancer of RNA stability. In contrast, HNRNPD promotes the degradation of its regulon, which results in an increase in their stability in AD [70,72]. The ability of pyPAGE to capture known and novel RBP associations with AD not only confirms the utility of this approach, but also provides readily testable hypotheses around the underlying modes of regulation.

To show relevance of the inferred associations in the context of AD, we explored biological functions of the identified RBPs. First, we looked at the mechanisms they use to control RNA metabolism (**Fig 5B**). Based on existing scientific literature, most of our candidate RBPs seem to regulate RNA stability directly upon binding either through repelling or recruiting RNA degradation factors [73–81]. However, some of these RBPs employ other mechanisms, such as HNRNPK which stabilizes RNA by preventing inclusion of cryptic exons during alternative splicing–a process that is critical in several neurodegenerative diseases [82]. Further, we performed analysis of overrepresented biological pathways within regulons of the identified RBPs to characterize their role in biological processes (**S8A Fig** and **S1 Table**). This analysis revealed

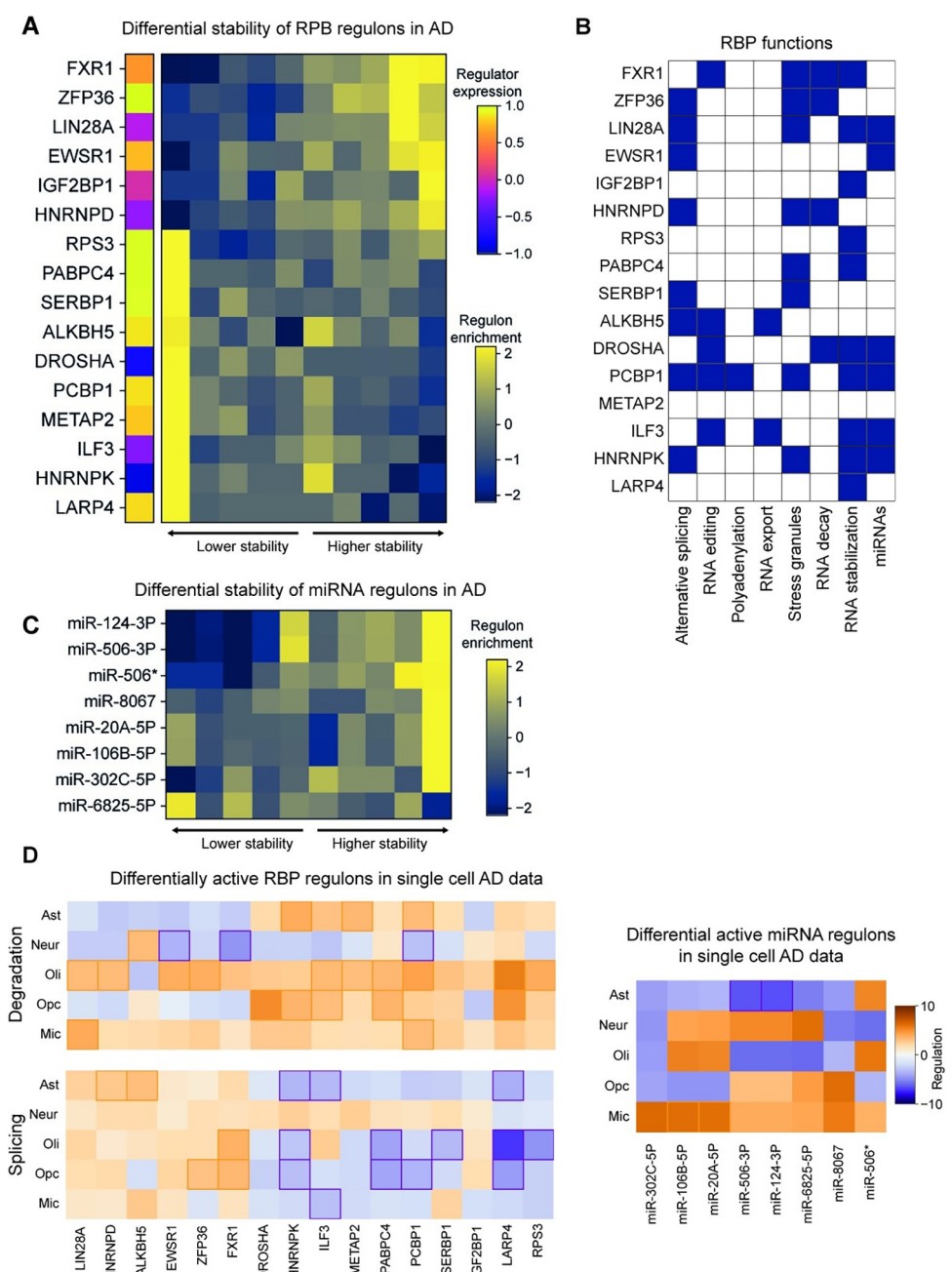

**Fig 5. Deregulation of post-transcriptional regulatory programs in AD. (A)** Heatmap representation of RBP regulons that are differentially expressed between AD and non-AD which we identified using pyPAGE. Here rows correspond to RBPs and columns to gene bins of equal size ordered by differential stability, the cells are colored according to the enrichment of genes from regulons in a corresponding bin. The leftmost column of this heatmap represents the differential expression of RBPs themselves. **(B)** Various roles performed by the identified RBPs based on the analysis of scientific literature. In this representation colored cells represent a recorded association between a protein and corresponding mechanism of action. **(C)** Deregulation patterns of the miRNA target gene-sets identified by pyPAGE. *miR-506 targets with GTGCCTT in their 3' untranslated region. **(D)** Differential activity of RBP and miRNA regulons in various brain cell types. The codes for the analyzed cell-types: neurons (Neur), astrocytes (Ast), oligodendrocytes (Oli), oligodendrocyte progenitor cells (Opc), microglia (Mic). Differential activity of RBP regulons was estimated based on differential rates of RNA splicing and degradation. miRNA regulons were analyzed using only estimates of degradation rates. In these heatmaps significant associations (p-value<0.05) are marked by colored frames. Regulation pattern is estimated as normalized conditional mutual information of the association multiplied by the sign of log fold change.

that these RBPs play a crucial role in regulating stability of various transcriptional regulators and genes involved in molecular signaling, including those associated with G protein-coupled receptor activity and kinase interactions. Differential activity of these RBPs in AD might in part explain observed deregulation of transcriptional programs via direct or indirect interactions. Additionally, the analysis of pathway enrichment within RBP regulons showed that a handful of the identified RBPs such PABC4, DROSHA, RPS3 and LARP4 are involved in the regulation of oxidative pathways which might be the root cause of AD-related mitochondrial dysfunction [83] and metabolic deficiencies [84].

## Identification of post-transcriptional programs governed by microRNAs in Alzheimer's disease

MicroRNAs are another major class of post-transcriptional regulators that have been previously associated with AD pathology [85]. Though many miRNAs have been reported to change their expression in the context of AD [85], there has been no systematic effort to infer their activity from changes in expression of their downstream targets. We sought to fill in this gap by applying pyPAGE to analyze AD-related coordinated changes in RNA stability of all annotated sets of miRNA targets, and identified eight candidate miRNA regulons (**Fig 5C**). In most cases AD-specific deregulation patterns of miRNA target genes were characterized by an increase in their stability. This suggests reduced activity of the miRNAs, which is in line with the notion that AD neurons lose the ability to process miRNAs [86]. Six out of eight identified miRNAs were previously associated with AD and its pathological mechanisms [87–91]. Functional analysis of genes regulated by these miRNAs (**S8B Fig and S1 Table**) showed that they largely control neuronal processes (neurogenesis, differentiation and signaling). Taken together, the marked deregulation of the identified miRNAs combined with their potential roles in AD and/or brain function makes them an attractive list of targets for further focused research.

## Cell-type specific analysis of post-transcriptional regulatory programs in AD

Similar to our characterization of cell type specific patterns of transcriptional regulons, we next assessed the activity of RBP- and miRNA-driven regulons in each of the disease-relevant cell types. For this, we analyzed single cell RNA-seq data from Morabito et al. [21], which contains cells from both neuronal and glial lineages. Since RNA stability is difficult to estimate directly from single cell RNA-seq data, we used functions from the 'velovi' package [92] for inference of RNA splicing and degradation rates within each cell type and compared healthy and AD donors. These estimates present an unbiased view of post-transcriptional regulation by controlling for the effect of transcriptional activity. Given their known regulatory functions, we used both differential stability and splicing rates to test for differential RBP activity. For miRNAs, on the other hand, we focused primarily on differential stability rates.

This analysis confirmed that the post-transcriptional regulatory programs we had previously identified in bulk RNA-seq data were predominantly deregulated in cells of glial lineage, and to a lesser degree in neurons (**Fig 5D**). Oligodendrocytes, astrocytes, and oligodendrocyte progenitors predominantly showed an increase in splicing rates and a decrease in degradation rates of RBP regulons in AD samples compared to controls. Neurons, on the other hand, did not exhibit broad changes in splicing rates and only a few RBP regulons showed changes in their degradation rates. Results from miRNA regulons, however, were less concordant between the single-cell and bulk data modalities. Four out of eight miRNAs we identified above showed significant deregulation patterns, and only in two cell types. We observed a significant decrease

in the stability of miR-106b-5p and miR-20A-5p regulons in microglia. Interestingly, upregulation of miR-106b-5p marks microglial activation [93], suggesting that its potential role as a regulator of inflammatory response in AD. Astrocytes showed a significant increase in the stability of miR-124-3p and miR-506-3p regulons, which are known to be critical for astrocyte transdifferentiation into neurons [94] and neuronal proliferation [95], respectively. In line with the importance of glia in AD pathology, overall our data highlights that most prominent patterns of miRNA and RBP regulon perturbations occur in non-neuronal cells.

## pyPAGE identifies deregulation of post-transcriptional programs as a major source of heterogeneity within the cohort of Alzheimer's patients

Finally, we employed pyPAGE to describe clinical implications of deregulation of the previously discovered master regulators. For that, we performed an analysis of their activity in each individual sample, measured using conditional mutual information (CMI) values that are provided by pyPAGE as part of its output. Here, the information criterion was estimated between the target genes and the standardized gene expression/transcript stability profile of each sample. As mentioned earlier, one of the benefits of using CMI is that it both captures monotonous changes in regulon abundance (**S9A Fig**) as well as non-linear patterns, which allows pyPAGE to efficiently measure regulon activity in individual samples.

Based on estimated activities of transcriptional and post-transcriptional regulons, we used Cox regression to assess associations between regulon activities and patient survival. Interestingly, we found that post-transcriptional programs are most strongly associated with patient survival, represented mainly by RBP regulons (LIN28A, HNRNPD, EWSR1, FXR1, HNRNPK, ILF3, PCBP1) and a few miRNA regulons (miR-302C-5P and miR-6825-5P) (**S9C Fig**). Using activity estimates of these RBP regulons, we performed unsupervised clustering of samples. This resulted in two clusters (**Fig 6A**), showing starkly different levels of activity of several RBPs. Remarkably, in this stratification low activity of the selected RBP regulons was associated with shorter survival (p-value = 0.007 of the Kaplan–Meier (log rank) test (**Fig 6B**) and hazard ratio of 4.84 (p-value = 0.012) of the Multivariate Cox model (**Fig 6D**)). In comparison, no other major biological covariate, such as age, sex and cognitive state, had a significant effect on survival (**Fig 6D**). Next, we established that the activity estimates of RBPs in samples from a group with shorter survival was similar to those observed in healthy samples (**Fig 6C**). This observation indicates that modulation of these post-transcriptional programs is likely a compensatory mechanism that helps mitigate effects of AD progression. Following this analysis we asked whether the samples from the group with low RBP activity can be further stratified based on the activity of the target RBPs. In a new stratification of these samples we found two sub-clusters, among which the one with higher activity showed increased survival as compared to the subcuster with lower RBP activity (p-value = 0.037 of the Kaplan–Meier test) (**S9D Fig**). These results highlight the underexplored significance of post-transcriptional regulation in AD and their potential clinical utility in improving patient stratification.

In order to identify underlying genetic root causes of the observed heterogeneity among AD patients, we performed differential analysis of genomic variants enrichment between the two identified groups of patients (**Table A in S1 Text**). Specifically, we looked at the AD-related GWAS [96] polymorphisms and polymorphisms related to genes of the analyzed RBPs. In this manner we have identified that patients with a longer lifespan are characterized by AD-related genomic variants in the upstream regions of CD55, PLK2 and SAP18 protein-coding genes. On the other hand, patients in the second group are characterized by variants upstream of STC1 and SCARA3 genes. The analysis of mutations associated with the RBPs showed that longer lifespan is associated with an upstream variant of PABPC4 gene and conversely shorter

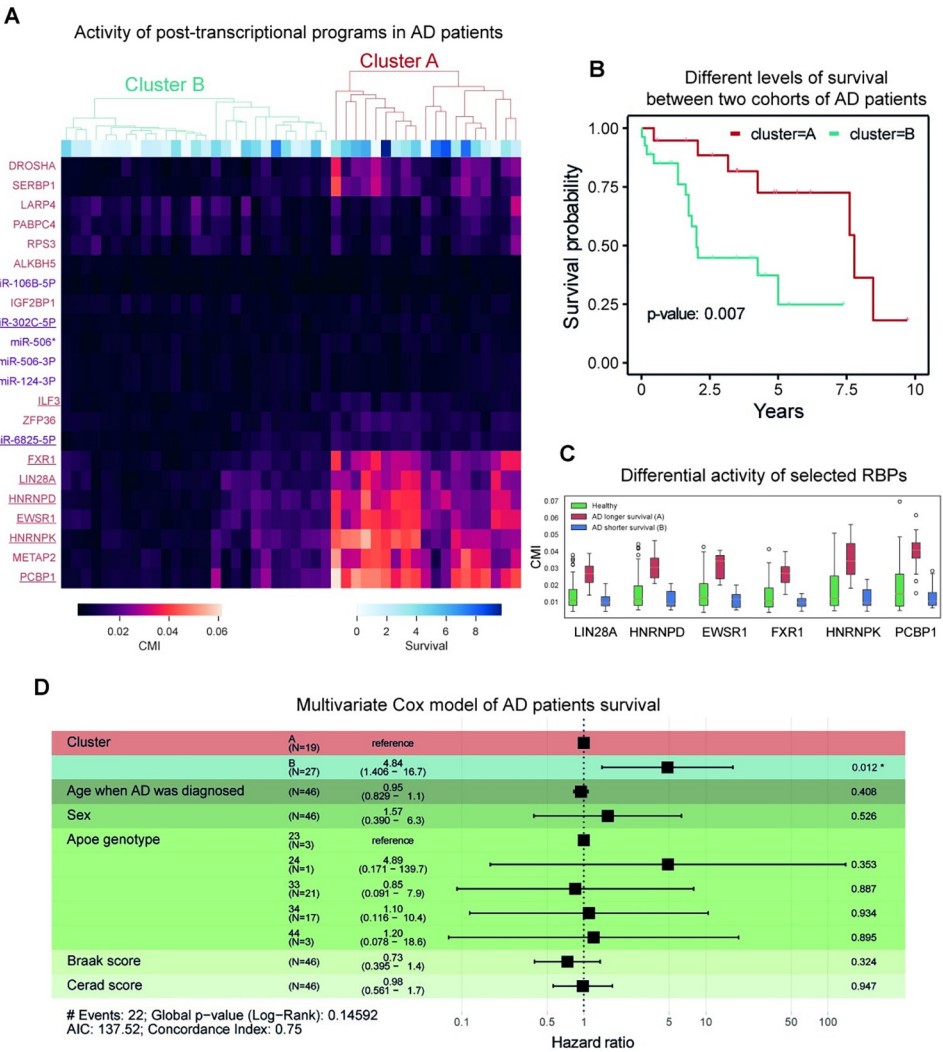

**Fig 6. Association of activation of post-transcriptional regulation programs with survival of patients with AD. (A)** Heatmap representing differences in the activity of the previously identified post-transcriptional regulons in AD samples. Factors which activity is significantly associated with survival are underscored. The dendrogram reflects the results of unsupervised clustering of samples based on activity of factors associated with survival. **(B)** Kaplan-Meier curve representing the difference in survival between two groups of patients stratified based on the activity of post-transcriptional regulons. **(C)** Comparison of the activity of selected RBP regulons in healthy samples and samples from two AD clusters. **(D)** Summary of the Cox regression analysis.

lifespan is related to two intronic variants in FXR1 and LARP4 genes. Incidentally, among these RBPs only FXR1 shows differential activity patterns between samples which makes it a primary subject for further research of the proteins which influence AD survival time.

## Discussion

Understanding the patterns of gene expression regulation in a biological system is a challenging task, but one which holds the potential to provide valuable insights into key factors controlling the behavior of a biological system. In this paper, we propose a systematic approach for exploration of gene regulation based on the analysis of regulon activity, which we implemented in our framework called pyPAGE. This new framework addresses two major problems

that complicate our understanding of gene expression: technical variation and the combinatorial action of multiple regulatory factors. First, by focusing on the differential activity of regulons rather than individual genes, pyPAGE becomes more robust to random variations in data. Second, by conditioning the association of gene expression with a particular regulon on the number of regulons each gene is associated with, pyPAGE effectively uses the structure of an interaction network to more efficiently disentangle perturbation responses of multiple factors. In settings with simulated data, pyPAGE outperforms other commonly used tools and demonstrates its robustness to noise in the input, such as data sparsity. We show that pyPAGE can be used for inference of changes in regulatory programs both on transcriptional and post-transcriptional levels and in bulk and single-cell data modalities.

In this study, we have showcased applications of pyPAGE to large RNA-seq datasets, both single-cell and bulk, in the context of Alzheimer's disease. We were able to recapitulate a number of previously known regulons and uncover new insights into the potential role of multiple TFs, RBPs and miRNAs in AD. Deregulation of the target genes controlled by these regulatory programs might explain pathological changes in AD, as they control key biological processes such as cell cycle progression, cellular signaling, neurogenesis, proteostasis and mitochondrial functions. By using single cell data of healthy and AD samples, we were able to capture the cell type specific patterns of the identified regulon that otherwise remained obscure in the analysis of bulk data. We found that AD-associated deregulations in gene expression in neurons are modulated primarily at the transcriptional level, whereas glial gene expression reprogramming is predominantly due to aberrant activity of post-transcriptional regulators. We further leveraged pyPAGE for the inference of regulatory programs hidden in the bulk data and uncovered characteristic upregulation of developmental TF regulons in most brain cell types. Similarly, we have identified that miR-124-3p, one of the key miRNAs that is depleted in AD brains, most strongly affects astrocytes, potentially blocking their transdifferentiation into neurons [94].

Like most complex human diseases, AD is a heterogeneous disease, likely with multiple molecular subtypes. Our results indicate that the activity of the various regulatory programs identified can be used to risk-stratify AD patients. In our stratification, patients with higher activity of several RBPs, in particular those associated with oxidative pathways, showed overall better clinical outcomes. As new therapeutic strategies against AD emerge, knowledge of these regulatory programs may allow for better stratification of responders and non-responders among the patient population.

## Materials and methods

### Analysis of bias in gene-set annotations

To demonstrate specific properties of gene-set membership distribution, we analyzed several commonly used annotations of gene-sets. Specifically, we examined annotations of biological pathways and miRNA targets [38] from MSigDB [3,97], TF regulons from Vorontsov et al. [37] and RBP regulons which we compiled ourselves. For each gene in an annotation we, first, calculated the number of gene-sets each gene is a member of, then, divided these values into several equally sized bins and then showed that this distribution can be closely approximated with the power law. In each case we report the inferred gamma parameter of the distribution and the $R^2$ statistics. We observed that for several annotations the long tail of the gene-set membership distribution deviates from the power law, indicating bias in annotations.

We further explored the possible sources of bias observed in annotations. Firstly, we analyzed the association between the number of protein interactions and the protein abundance. To determine the number of interactions, we utilized the STRING database [98] considering

only physical interactions with a confidence score greater than 0.9. As a source of protein abundance data we used proteome measurements from HEK-293 cells [99]. Secondly, we analyzed the association between pathway membership and citation index [100] for each gene in the annotation of biological pathways. In both cases we were able to show that there is a linear relationship between the analyzed variables indicated by high correlation values.

## pyPAGE algorithm

The computational framework implemented by pyPAGE includes three key steps. First, it preprocesses the input data by discretizing the differential gene expression profile into equally sized bins and converting a gene-set annotation into a binary format. Second, it performs estimation of conditional mutual information between differential gene expression and each gene-set. Here the metric is conditioned on the gene-set membership profile which is also discretized into several bins.

Formula for calculating conditional mutual information is presented below:

$$I(X;Y|Z) = \sum_{z \in Z} p_Z(z) \sum_{y \in Y} \sum_{x \in X} p_{X,Y|Z}(x,y|z) log \frac{p_{X,Y|Z}(x,y|z)}{p_{X|Z}(x|z)p_{Y|Z}(y|z)} \quad (1)$$

In our algorithm, X stands for a set of bins in the expression vector (by default 10 bins), Y for a set {0, 1} which represents presence or absence of a gene in a regulon and Z is a set of bins (by default its size is 3) into which we discretize gene-set membership. Prior to computation of the metric, no probability smoothing is performed. In order to speed up our method, we implemented an efficient computation of the metric using numba package [101].

Following estimation of conditional mutual information, the program estimates the significance of the inferred associations. For this, a non-parametric testing procedure is implemented. The differential expression profile is randomly shuffled multiple times and at each iteration the conditional mutual information is calculated between these profiles and the scrutinized gene-set. Then the significance of the association is inferred as the proportion of cases when the CMI of randomly permuted differential expression profile turned out to be greater than CMI of the original profile. This procedure is applied to a list of gene-set in a descending order of the CMI calculated at the second step. When multiple insignificant associations (by default 5) come in a row, the testing stops.

pyPAGE may also use an additional filtering procedure which is applied from top-to-bottom to the list of most informative gene-sets. Each next gene-set is required to fulfill following requirement:

$$\frac{I(candidate\_gene\_set;\ expression|accepted\_gene\_set)}{I(candidate\_gene\_set;\ accepted\_gene\_set)} > r \quad (2)$$

r represents an increase in the amount of information and its optimal values typically lie between 0 and 0.5 (by default it is set to 0.1). The idea behind this procedure is that in the final output should be a representative set of gene programs (regulons), where each new entry adds a significant amount of information to the information that is already present in the selected ones. It is important to note that this filtering is not related to annotational biases and so can not replace estimation of conditional mutual information in the first place.

The final output of pyPAGE consists of gene-sets that, first, exhibit significant association with expression represented by high CMI and, second, pass a redundancy test. The resulting list can be additionally filtered using multiple testing correction methods. However, in practice we observe that all gene-sets that pass previous tests pass the Benjamini–Hochberg procedure. In the end results are reported in the form of a table. It also has an option of the graphical

output, which is a heatmap representing gene-set members enrichment within equally sized bins ordered by differential gene expression. To quantify the level of over- and under-representation in each bin, we employ the hypergeometric test to calculate two distinct p-values: one for the likelihood of observing fewer genes from the same gene-set in a given bin (under-representation), and another for the probability of observing more genes (over-representation). If the p-value for over-representation is lower than that for under-representation, we consider the gene-set to be over-represented in that bin; otherwise, it is considered under-represented. The minimal p-value is then log-transformed, and multiplied by -1 if the gene-set is over-represented.

pyPAGE is implemented as a python package and is openly available at https://github.com/goodarzilab/pypage.

## Tests

In order to validate various aspects of pyPAGE execution we have developed a series of tests based on simulated data. Our main focus was on proving that the program is robust to gene-set annotation biases. To do that we evaluated its average performance using artificial datasets and benchmarked it against other programs. We also conducted two additional tests to demonstrate consistency of pyPAGE predictions. Though to produce highest performance pyPAGE needs to receive clean data with strongly manifesting biological signals, we still expected the program to perform decently in various adversarial cases. By subsampling the original input and by modulating the robustness of signals encoded in the data, we wanted to show at which cost the reduction in input quality is to the overall performance. The full description of the tests is provided further.

## Simulating differential expression

All implemented tests required simulating differential gene expression data. Though in principle pyPAGE does not depend on any particular type of the input distribution, we decided to make the input data realistic. So, we based the simulation on the actual gene expression estimates from a real biological system, which in this case was activation of murine CD4 T cells (PRJNA296380) [102]. Starting off with fastq files, we first applied Salmon [103] to quantify gene counts and afterwards applied DESeq2 [68] to estimate differential gene expression between two groups of samples. To produce an arbitrary amount of simulated gene expression profiles using this data, we performed random shuffling of the data across genes. This procedure minimized the informational content which had been in the data initially. We demonstrated that by performing pyPAGE gene-set enrichment analysis of terms from biological pathway annotation using original and augmented data. While the analysis of the original data reported 76 pathways, using simulated data pyPAGE on average reported only 3.36 pathways (standard deviation = 1.80). Though this procedure removes almost completely any informational content, it preserves the properties of the original data distribution. Moreover it can be repeated multiple times to produce many samples of simulated differential expression profiles.

## Simulating pathway annotations

In addition to the differential expression profiles, each test needed a simulated pathway annotation. To be able to assess the performance of pyPAGE at later stages, during annotation design whether a gene-set is deregulated or not should be directly encoded in its structure. And so our goal was to create annotations each containing in total 1100 gene-sets, among which 100 are known to be deregulated in the corresponding differential gene-expression profile. The number of genes that went into a newly designed gene-set chosen based on the

distribution of sizes of TF regulons [37]. To simulate unperturbed gene-sets we randomly picked the necessary number of genes. To imitate deregulated gene-sets, we used genes that are either over- or under-expressed in the corresponding simulated data; the details of simulating such gene-sets are described in the following paragraph.

When simulating deregulation our objective was to imitate patterns observed in real systems, so as to avoid making the task of predicting gene-set deregulation too easy or too difficult. In a real case scenario genes of a deregulated gene-set are not always exclusively found among the most overexpressed or underexpressed genes, rather the signal is distributed over the whole range of values. Likewise, in our simulation we built probabilistic models to control which genes go into a newly formed gene-set. We aimed that the resulting distribution of differential gene expression in each gene-set should match the truncated normal distribution on an interval between -1 and 1 with predefined parameters. So for each gene-set we first defined a distribution of expected differential gene expression, then randomly sampled a number of values from it and selected those genes from a simulated differential expression profile which most closely matched these values. Parameters of the expected distribution were also selected randomly. For an upregulated gene-set, its mean was picked randomly from a random distribution in an interval between 0.5 and 1, for a downregulated gene-set the interval was between -1 and -0.5. The standard deviation was uniformly selected in the range between 0.1 and 0.2. To simulate dual regulation, we mixed genes selected for an upregulated gene-set with those of a deregulated one. Finally, to imitate biological and technical noise to each gene-set we added a randomly selected group of genes. The ratio of randomly added genes was controlled by signal-to-noise ratio, which in the general case was set to 33%.

The resulting gene-set annotations were unbiased with gene-set membership following approximately normal distribution. To produce biased annotations, we needed to add bias externally. And so we selected 2000 genes which would represent genes with promiscuous activity. Then each gene-set in an annotation was concatenated with additional 100 genes randomly selected from this group. The idea behind this approach is that added genes would occur much more frequently than all other genes effectively creating bias in gene-set membership distribution.

## Benchmarking pyPAGE

Using the artificial datasets containing simulated differential expression and gene-set annotations, we were able to analyze pyPAGE effectiveness and compare it with other programs: iPAGE and fgsea. The test was basically a classification task: each program needed to predict deregulated gene-sets. The performance was judged based on the knowledge of true patterns of regulation which were encoded in simulated annotation. We used multiple metrics to assess performance of programs which were averaged across 20 iterations with different sets of simulated data. To present these results we plotted ROC (receiver operating characteristic) curves, PR (precision recall) curves, DET (detection error tradeoff) curves, FDR (false discovery rate curves) curves and calculated respective AUC (area under the curve) scores. Since the number of unperturbed gene-sets was much greater the most relevant metric in this task was PR-AUC score. Programs were tested in 4 different conditions to reflect the effect of bias and non-linear regulation on the quality of predictions.

## Robustness test

Since in practical applications, the input data of a gene-set can often be corrupted, we developed two tests to explore how robust pyPAGE is to the augmentation of the input. The first strategy that we implemented for simulating input perturbation was increasing the amount of

noise in the data. In the previous we set signal-to-noise ratio to 33%, however here we varied this parameter in the range between 0 and 100%. The second type of augmentation we used was implemented by randomly subsampling the input data. We iteratively decreased the number of genes in the input by 5% measuring pyPAGE performance. Both tests output pyPAGE sensitivity changes in response to input modulation.

## Analysis of Alzheimer's disease

To characterize perturbation of transcriptional and post-transcriptional programs in AD, we analyzed bulk RNA-sequencing dataset from MSBB study [18]. We focused on samples of BM36 brain region and considered only those that either were annotated as definitely pathological or normal, which left 163 samples. After that we processed the input fastq files with Salmon [103] to quantify gene counts, which we analyzed using DESeq2 [24]. As part of DESeq2 analysis, we performed principal component analysis, which showed that samples cluster within 4 groups corresponding to a sex of the donor and a batch group. Each of these groups contained both healthy and pathological samples, however one of the batch groups showed poor separation between healthy and pathological samples and so we excluded them from our analysis. For the remaining 97 samples we estimated fold change of each gene and corresponding p-values, doing so separately for male (35 samples) and female (62 samples). In order to account for any sex-specific variation, we combined estimated p-values using a modified inverse chi-square method for correlated tests [104] as well as values of log fold change using mean. Afterwards we quantified differential gene expression using the following formula:

$$DE = (1 - p\_value) \cdot sign(log(fold\_change)) \tag{3}$$

Finally, we used these estimates to perform pyPAGE analysis of TF regulons deregulation using gene-sets from Vorontsov et al. [37] (full description of the annotation is provided in **Table B in S1 Text**). For this run we used the following set of parameter values: alpha 0.01, redundancy ratio 0.2, number of expression bins is 10 and the number of bins into which the gene-set membership vector is split is 3. The overall summary of this and following pyPAGE experiments can be found in **Table C in S1 Text**.

In order to analyze post-transcriptional regulation, we first needed to estimate differential transcripts stability. To do this we first applied Salmon [103] to quantify separately exonic and intronic counts and then applied REMBRANDTS [27] to produce unbiased estimates of stability for each sample. Next using t-test we compared stability between pathological and normal samples; the resulting p-values and fold changes were used to quantify differential stability similarly to how we estimated differential gene expression. Finally, based on these estimates pyPAGE was able to infer deregulation patterns of RBP regulons from our newly compiled annotation and microRNA targets [38] from MSigDB [3,97] (**Table C in S1 Text**). pyPAGE analysis of RBPs was done with alpha set to 0.05, redundancy ratio 0.3, number of expression bins 10, gene-set membership bins 3. miRNAs were analyzed using number of expression bins 10, gene-set membership bins 3, alpha 0.01 and no redundancy filtering, instead we selected only those miRNAs whose targets were significantly enriched either among top differentially upregulated genes or top differentially downregulated genes. This was done because we wanted to limit our analysis to the strongest signals in the data.

Cell-type specific deregulation patterns were estimated using single cell data. For this purpose we used two datasets. In both cases we relied on prior annotation of cell-types provided by the authors of the studies. One was an already processed dataset presented by the ROSMAP study [20]. The dataset contained 70634 cells from 48 donors and was balanced in respect to

pathology and sex covariates. We used it for the analysis of TF regulons. There within each cellular type cluster we applied t-test to estimate differential gene expression between cells from AD and non-AD samples. Then pyPAGE was used to estimate deregulation of TF regulons within each cell type. To analyze activity of the regulators that we previously detected in our analysis we set redundancy ratio to 0.0 and alpha to 1.0, this prevented pyPAGE from truncating its input only to significant associations. In addition we also ran pyPAGE with redundancy ratio being set to 0.3 and alpha to 0.05 which allowed us to identify TFs whose perturbation remained hidden in the bulk data. For the analysis of RBP and miRNA deregulation, we used another single cell RNA-seq dataset from Morabito et al. [21]. In this data there were 18 samples in total of which 11 were from AD patients (6 female, 5 male) and 7 from healthy donors (2 female, 5 male). In total there were 61472 cells. We processed this dataset de novo using cellranger [105] and velocyto [106] in order to finally run velovi [92]. We applied velovi to single cell datasets of individual samples in order to estimate patient specific rates of RNA splicing and degradation in each cellular type. In the end in a manner similar to how we estimated differential gene expression, here we used t-test to estimate differential kinetic rates between patients with and without AD. Based on the differential splicing and degradation rates, using pyPAGE we were able to infer the activity of RBP regulons. To analyze miRNA regulons, we used only differential degradation rates. With RBPs and miRNAs we only wanted to get estimates of differential activity of the factors identified in bulk analysis, so in all of the cases we set redundancy ratio to 0.0 and alpha to 1.0.

Spatial analysis of TF deregulation patterns was performed using sequencing data from the Visium platform of the middle temporal gyrus [22]. In total the dataset had 25293 (22085 after filtering out ones with less than 2000 gene counts) spots from 6 cortical slices (3 healthy and 3 AD). The count data was scaled and then log normalized after which we computed differential gene expression between AD and control samples using the Wilcoxon test. This was done for each cortical layer relying on their prior annotation. Further, we applied pyPAGE to estimate differential activity of TFs that we previously identified as associated with AD. In the end for each TF in each cortical layer we report its deregulation pattern computed as normalized CMI multiplied by the sign of log fold change and its significance.

We also analyzed the association between the activity of the identified regulons and patients' survival. To do this, we analyzed another bulk RNA-seq dataset from the ROSMAP study [107] stored in the Synapse database under syn21589959. The samples were from dorsolateral prefrontal cortex and posterior cingulate cortex. We considered only samples from AD patients for whom the age the disease was first diagnosed was not censored. This left us with 46 samples. For these samples we performed gene expression quantification using salmon and then normalized these estimates. We then used this normalized gene expression to quantify activity of TF regulons in each sample using pyPAGE estimates of conditional mutual information. Similarly, we used salmon and REMBRANDTS to estimate transcripts stability in each sample and afterwards used normalized stability to infer activity of RBP and miRNA regulons. Activity of regulons was estimated on a per-sample basis, so that instead of differential expression/stability, pyPAGE was run with normalized expression/stability values for each sample. Next, we used regulon activity in Cox Proportional-Hazards Model to identify factors associated with survival of AD patients. Based on this we selected regulons which were significantly associated with survival. We then analyzed patterns of activity of these regulons in various samples and effectively excluded those regulons which did not show any marked difference in their activity between different samples. Then based on the activity of the remaining regulons we performed clustering of samples which resulted in two big clusters. Next, we used this stratification to analyze its association with survival. For this, we built Kaplan-Meier curve and

performed Cox Proportional-Hazards Model which incorporated other AD relevant covariates such as sex, APOE genotype, Braak score, Cerad score.

For the two clusters that we identified in our survival analysis we looked at the polymorphisms that characterize these groups using a complementary WGS dataset (syn2580853). In particular we considered 2 categories: those that are associated with AD polymorphisms from GWAS catalog [96] and those that are associated with the RBPs from our analysis. We report variants that are enriched in one group of samples (present in over 80% of samples) or depleted in another (present in fewer than 20% of samples).

### Characterization of regulon functions

We describe the functions of the identified TF, RBP and miRNA regulons based on the enrichment of molecular pathways among their members. To do that, we used a statistical overrepresentation test from PANTHER [108], which includes multiple testing corrections. This test estimated the significance of pathway enrichment within a set of all genes that are targets of a specific factor. Then we aggregated the inferred functions of each regulon and presented them in a heatmap form.

For RBPs we also characterize their roles in various RNA metabolic processes. This was done based on manual curation of scientific literature. We describe RBP associations with following molecular mechanisms: alternative splicing [109–116], RNA editing [117–119], polyadenylation [120], RNA export [121,122], stress granule formation [123,124,124–129], RNA-decay [72,130–132], RNA stabilization [73–81] and miRNA functions [132–137].

### ENCODE processing

The ENCODE eCLIP data [67] was available for two cell lines: HepG2 and K562. In total, there were 234 <RBP, cell type> pairs (687 samples including replicates, and matched controls for each RBP). RBPs with no matched control or with matched control only (without the experiments) were excluded from the analysis resulting in the final set of 222 <RBP, cell type> pairs. The data were analyzed as follows: (1) reads were preprocessed as in the original eCLIP pipeline [138], (2) trimmed reads were mapped to the hg38 genome assembly using hisat2 [139], (3) the aligned reads were deduplicated [138], (4) the uniquely and correctly paired reads were filtered with samtools [140], (5) gene-level read counts in exons were obtained with plastid [141] using gencode v31 comprehensive gene annotation [142], therefore joint read counts were obtained for genes with overlapping annotations, (6) differential expression analysis against matched controls was performed with edgeR [26], (7) Z-Scores were estimated for each gene across data for all RBPs for the fold change values estimated in (6). Based on (6) and (7) reliable RNA targets of each RBP were defined as those passing 5% FDR, raw log2FC > 0.5, Z-Score(log2FC) > 1.645. The processed data is stored in S2 and S3 Tables.

### Supporting information

**S1 Fig. Bias in gene-set annotations. (A)** Characteristics of a gene-set membership degree distribution in the TF regulons annotation. The top plot represents the observed distribution with a power law function being fit to it. We also report the gamma parameter of this distribution and the $R^2$ The bottom plot represents the deviation of the observed distribution from the power law. **(B)** Similar representation for RBP regulons. **(C)** Similar representation for miRNA targets. **(D)** Similar representation for GO pathways.
(TIFF)

**S2 Fig. Performance of pyPAGE benchmarked against iPAGE and fgsea in various simulated conditions.** For comparison we used multiple metrics, results are presented as ROC, PR, DET curves and plot of dependency between alpha and FDR.
(TIFF)

**S3 Fig. Variation in expression of TF regulons.** **(A)** Boxplots representing expression of TF regulons which were identified using pyPAGE in AD and non-AD samples from female and male donors. **(B)** Boxplots representing expression of the same TF regulons in AD and non-AD samples in different age cohorts.
(TIFF)

**S4 Fig. Top 100 of pathways enriched in TF regulons.** **(A)** Heatmap representing biological pathways that are enriched among targets of TFs we have identified in our analysis. Here we present top 50 pathways which are significantly enriched in most of the regulons excluding several generic terms, like "molecular process". **(B)** This heatmap represents another 50 pathways significantly associated with the analyzed TFs.
(TIFF)

**S5 Fig. Cell-type specific patterns of TF target genes deregulation in AD.** **(A)** Differential activity of TFs in excitatory neurons inferred using pyPAGE based on the analysis of single-cell RNA-seq data. **(B)** TF regulons deregulated in inhibitory neurons. **(C)** TF regulons deregulated in astrocytes. **(D)** TF regulons deregulated in oligodendrocyte progenitors. **(E)** TF regulons deregulated in microglia. **(F)** TF regulons deregulated in oligodendrocytes. **(G)** TF regulons deregulated in endotheliocytes. **(H)** TF regulons deregulated in pericytes.
(TIFF)

**S6 Fig. Correlation between the expression of RBPs and their regulons' average stability.** The barplot representing the correlations between the expression of RBPs and the average stability of their regulons. Asterix indicated significant correlation (p-value<0.05).
(TIFF)

**S7 Fig. Variation in expression of TF regulons.** **(A)** Boxplots representing stability of RBP regulons which were identified using pyPAGE in AD and non-AD samples from female and male donors. **(B)** Boxplots representing stability of the same RBP regulons in AD and non-AD samples in different age cohorts. **(C)** Boxplots representing stability of miRNA targets within AD and non-AD samples from female and male donors. **(D)** Boxplots representing stability of miRNA targets within AD and non-AD samples from different age cohorts.
(TIFF)

**S8 Fig. Top 100 pathways enriched in RBP and miRNA regulons.** **(A)** Heatmap representing biological pathways that are enriched among targets of TFs we have identified in our analysis. Here we present only the top 100 pathways which are significantly enriched in most of the regulons. **(B)** Similar representation for miRNA targets.
(TIFF)

**S9 Fig. Association of expression of a subset of HOX genes with survival of patients with AD.** **(A)** Comparison of the similarity between various regulon activity metrics, namely conditional mutual information (CMI), mean regulon abundance and the expression of a factor itself. **(B)** Heatmap representation of activity of transcriptional regulons in different patients. **(C)** Bar Plot representing significance of association of activity of each previously identified regulon with patient's survival. **(D)** Kaplan-Meier curve representing the difference in survival between two groups of patients stratified based on the activity of post-transcriptional regulons

within the group with low RBP regulon activity.
(TIFF)

**S1 Text.** Table A in S1 Text. Differential enrichment of genomic variants in clusters identified based on RBP activity. Table B in S1 Text. Description of gene-set annotations. Table C in S1 Text. Summary of the pyPAGE experiments.
(DOCX)

**S1 Table. Functions of the identified regulators.**
(XLSX)

**S2 Table. Processed ENCODE K562 eCLIP data.**
(TSV)

**S3 Table. Processed ENCODE HepG2 eCLIP data.**
(TSV)

## Acknowledgments

We thank Andrey Buyan and Ivan V. Kulakovskiy for providing reprocessed ENCODE eCLIP data. We also thank Dr. April Pawluk for reviewing and editing this paper.

The results published here are in whole or in part based on data obtained from the AD Knowledge Portal (https://adknowledgeportal.org/). These data were generated from postmortem brain tissue collected through the Mount Sinai VA Medical Center Brain Bank and were provided by Dr. Eric Schadt from Mount Sinai School of Medicine.

## Author Contributions

**Conceptualization:** Artemy Bakulin, Martin Kampmann, Matvei Khoroshkin, Hani Goodarzi.

**Data curation:** Artemy Bakulin.

**Investigation:** Artemy Bakulin.

**Methodology:** Artemy Bakulin, Matvei Khoroshkin.

**Software:** Artemy Bakulin, Noam B. Teyssier, Matvei Khoroshkin.

**Supervision:** Hani Goodarzi.

**Validation:** Artemy Bakulin, Matvei Khoroshkin.

**Visualization:** Artemy Bakulin.

**Writing – original draft:** Artemy Bakulin, Matvei Khoroshkin, Hani Goodarzi.

**Writing – review & editing:** Artemy Bakulin, Martin Kampmann, Hani Goodarzi.

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
