## [Decision Letter · Decision Letter 0]

13 Nov 2023

Dear Dr Goodarzi,

Thank you very much for submitting your manuscript "Addressing biases in gene-set enrichment analysis: a case study of Alzheimer’s Disease" for consideration at PLOS Computational Biology.

As with all papers reviewed by the journal, your manuscript was reviewed by members of the editorial board and by several independent reviewers. In light of the reviews (below this email), we would like to invite the resubmission of a significantly-revised version that takes into account the reviewers' comments.

We cannot make any decision about publication until we have seen the revised manuscript and your response to the reviewers' comments. Your revised manuscript is also likely to be sent to reviewers for further evaluation.

Sincerely,

Andrey Rzhetsky

Academic Editor

PLOS Computational Biology

Pedro Mendes

Section Editor

PLOS Computational Biology

Reviewer's Responses to Questions

**Comments to the Authors:**

Reviewer #1: Uploaded as attachment

Reviewer #2: uploaded as an attachment

Reviewer #3: Bakulin et al. present the novel pyPAGE method for identifying transcriptional and post-transcriptional regulators underlying a condition from expression, RNA stability, or splicing stability differential measures, and previously identified regulator targets (regulons). pyPAGE, as its predecessor, iPAGE, is based on mutual information between the differential measures and the target indicator for each of the studied transcriptional or post-transcriptional regulators, allowing for the detection of monotonic and non-monotonic effects of the regulator. pyPAGE differs from iPAGE in that it directly uses the biases of the regulator target annotations. Some genes are targets of many regulators, while a single regulator targets others, so a power law distribution often approximates the number of regulators per gene.

The authors first demonstrate the biases in a few annotation databases and provide evidence that differences in the protein abundance or study biases correlate with the observed features of the distributions of the numbers of regulators or pathways per gene. Then, they describe and evaluate the pyPAGE method on simulated datasets. The evaluation demonstrates that pyPAGE handles better annotation biases than iPAGE and is robust to input down sampling or decrease in the signal-to-noise.

The rest of the paper shows many applications of pyPAGE to analyze bulk and single-cell RNASeq datasets in the context of Alzheimer’s disease (AD). Transcription factors (TF) are studied using previously defined targets and differential expression (DE) measures (AD vs control samples) from the bulk RNASeq data from the BM36 region from the MSBB study and the single-cell RNASeq data from the ROSMAP study. RNA binding proteins (RBP) are studied using a collection of RPB targets compiled by the authors and differential RNA stability measurers from the bulk RNASeq MSBB datasets and differential splicing and deregulation measures from the single-cell RNASeq from Morabito et al. Finally, the authors examine microRNAs through microRNA targets from the MSigDB and DE from the bulk MSBB and differential splicing and deregulation from the single-cell Morabito et al.

Several regulators, TF, RBP, and microRNAs exhibit differences in their regulon’s expression measures in the bulk RNASeq data. The authors show that these are misregulated either in all cell types of the single-cell data or in one or more specific cell types. Moreover, the authors identified additional regulators through independent analysis of the cell types.

Finally, the authors split the AD patients from a different bulk RNASeq dataset from the ROSMAP study into two clusters based on the activity of a set of post-transcriptional regulators (RPB and microRNA). The two clusters differ significantly in the years of survival post-diagnosis. The regulators used for clustering were among the ones the authors identified as misregulated in the prior analysis. Several AD-related SNPs (by GWAS) were also found to have different frequencies in the two groups.

I’m in no position to evaluate the insights gained from the study of AD critically. Although I find the paper potentially interesting to the scientific community, I have a few concerns regarding the description of the pyPAGE, its evaluation, and its applications.

At its core, pyPAGE is based on conditional mutual information (CMI). I would suggest adding the definition/formula of CMI in the methods section. (As a minor issue, CMI is used in the text before defining it). Is any smoothing applied while estimating the probabilities and conditional probabilities used to compute CMI?

In the pyPAGE’s description in the Methods section (and only in the Methods), there is a discussion of the “additional filtering” procedure. I find this additional filtering intriguing. If I understand correctly, filtering is an alternative approach to CMI to address annotation biases where a gene is present in many regulators/pathways. The main problem with such genes is that, when differentially expressed, they may cause many regulators to appear misregulated.

pyPAGE addresses this source of false positives by down weighting the importance of the promiscuous genes through CMI and the number of regulators per gene. The filtering procedure addresses the issue of only considering regulators whose misregulation cannot be explained by previously accepted regulators. These thoughts bring me to the following list of suggestions:

• Expand the description of the filtering procedure by adding motivation.

• Clearly state if and where (evaluation, AD applications, etc.) it is used and with what r.

• Can the filtering procedure by itself ‘rescue’ iPAGE?

• Is the filtering procedure essential for the good performance of pyPAGE.

• If none of the above is interesting and important, remove the discussion of the filtering.

The generation of synthetic data that mimics the hypothesized effects of biases in the annotation database seems convoluted and arbitrary. But after spending a few days trying to suggest something better, I gave up. The described procedure does generate datasets that make the performance of the pyPAGE better than that of iPAGE. This is not surprising, as pyPAGE directly accounts for the simulated biases, but tt the very least, that shows that pyPAGE’s implementation is likely correct. I think the paper will have a much stronger argument if the authors also compare pyPAGE’s and iPAGE’s results on a real dataset. The expectation is that iPAGE will generate more significant regulators. It would be reassuring if the authors could show that the extra ones are artifacts caused by the annotation biases.

My main criticism is related to the description of the AD experiment performed.

Many annotation datasets of regulators and their targets and bulk and single-cell RNASeq datasets are used. Basic information about all these datasets is missing. For all annotation databases, it is essential to present the number of regulators, the number of targets, and the distributions of the number of regulators per gene. For the RNASeq datasets, the authors should include the number of samples, control, and cases in the original dataset, the one used for the analysis after filtering, and a brief explanation about why the dataset is used.

I find the pipeline shown in Figure 1 C misleading: (1) It does not reflect the presented experiments. (2) What is gene count? (3) It seems the pyPAGE jointly analyzes DE and differential transcript stability while these are analyzed independently. I’d suggest showing a diagram or a table listing all experiments discussed in the results. The table should list the annotation database, RNASeq data, and the measure used for each experiment.

It is also important to be specific about the parameters used for the analyses. How many bins are used? What is the r for the additional filtering? What are the detailed versions and parameters of the tools used to compute the differential measures from the RNASeq datasets?

There are several choices made that seem arbitrary and not well justified:

• It is unclear why the BM36 region is used for the bulk RNASeq analysis.

• Can the DE be used with the RBPs and microRNAs instead of the differential RNA stability?

• Why does the single-cell RNA analysis of the RPBs and microRNAs use differential splicing and deregulation measures instead of the differential RNA stability as done in the bulk RNASeq analysis?

• Why is a different dataset used to demonstrate the heterogeneity of AD patients?

I don’t quite understand how the pyPAGE’s CMI output is used to measure the activity of a regulon in an individual. As the split of the AD patients into two groups is a major result, the procedure should be made explicit in the methods.

Minor

• “… genes themselves, that underlies cellular pathologies” -> “… genes themselves, that underly cellular pathologies.”

• What does “and uses an underlying deterministic statistic rather than permutation-based approaches” mean? Aren’t the p-values in pyPAGE based on non-paramemtric permutation?

• What are the typical bin numbers used for differential expression and the gene set membership profiles?

• In the left column of Figure 2A, a direction should be indicated. I.e., yellow means the TF has an increased expression in AD. How are the enrichment scores for the bins computed?

• Figure 2E. Are the TF functions identified using only the subset of the regulon targets exhibiting DE or all targets?

• The legend of Figure 2 is not in sync with the figure. The panel labels are permuted.

• I would suggest splitting Figure 1: panels A and B, focusing on the properties of gene annotations and the rest, focusing on the pyPAGE and its evaluation.

• Tables S1 and S2 that summarize the main results of the AD analysis seem very useful. Can you promote it to the main text?

**Have the authors made all data and (if applicable) computational code underlying the findings in their manuscript fully available?**

Reviewer #1: Yes

Reviewer #2: Yes

Reviewer #3: Yes

PLOS authors have the option to publish the peer review history of their article (what does this mean?). If published, this will include your full peer review and any attached files.

Reviewer #1: No

Reviewer #2: No

Reviewer #3: No
---

## [Decision Letter · Decision Letter 1]

22 Jul 2024

Dear Dr Goodarzi,

We are pleased to inform you that your manuscript 'pyPAGE: A Framework for Addressing Biases in Gene-Set Enrichment Analysis — A Case Study on Alzheimer’s Disease' has been provisionally accepted for publication in PLOS Computational Biology.

Best regards,

Andrey Rzhetsky

Academic Editor

PLOS Computational Biology

Pedro Mendes

Section Editor

PLOS Computational Biology

Please take care of the minor changes requested. Once this is don, the manuscript will be accepted.

Reviewer's Responses to Questions

**Comments to the Authors:**

Reviewer #2: All my comments are addressed, except for the real-case comparisons and authors need to (1) explicitly describe the real-case problem setup such as introduction of real-case data; (2) there shall be real-case examples with ground-true labels that were confirmed by experiments or model predictions, and then assessing prediction accuracy, NPV, and PPV is more relevant than result consistency.

Reviewer #3: The authors have addressed my concerns well. I find the revised manuscript to be substantially improved.

**Have the authors made all data and (if applicable) computational code underlying the findings in their manuscript fully available?**

Reviewer #2: Yes

Reviewer #3: Yes

PLOS authors have the option to publish the peer review history of their article (what does this mean?). If published, this will include your full peer review and any attached files.

Reviewer #2: No

Reviewer #3: No

---

## [Editor Report · Acceptance letter]

28 Aug 2024

PCOMPBIOL-D-23-01403R1 

pyPAGE: A Framework for Addressing Biases in Gene-Set Enrichment Analysis — A Case Study on Alzheimer’s Disease

Dear Dr Goodarzi,

I am pleased to inform you that your manuscript has been formally accepted for publication in PLOS Computational Biology. Your manuscript is now with our production department and you will be notified of the publication date in due course.

With kind regards,

Anita Estes
